# Synthesis and Characterization of Novel Methyl (3)5-(*N*-Boc-piperidinyl)-1*H*-pyrazole-4-carboxylates

**DOI:** 10.3390/molecules26133808

**Published:** 2021-06-22

**Authors:** Gita Matulevičiūtė, Eglė Arbačiauskienė, Neringa Kleizienė, Vilija Kederienė, Greta Ragaitė, Miglė Dagilienė, Aurimas Bieliauskas, Vaida Milišiūnaitė, Frank A. Sløk, Algirdas Šačkus

**Affiliations:** 1Institute of Synthetic Chemistry, Kaunas University of Technology, K. Baršausko g. 59, LT-51423 Kaunas, Lithuania; gita.matuleviciute@ktu.lt (G.M.); neringa.kleiziene@ktu.lt (N.K.); greta.ragaite@ktu.lt (G.R.); migle.dagiliene@ktu.lt (M.D.); aurimas.bieliauskas@ktu.lt (A.B.); vaida.milisiunaite@ktu.lt (V.M.); 2Department of Organic Chemistry, Kaunas University of Technology, Radvilėnų pl. 19, LT-50254 Kaunas, Lithuania; vilija.kederiene@ktu.lt; 3Vipergen ApS, Gammel Kongevej 23A, V DK-1610 Copenhagen, Denmark; fas@vipergen.com

**Keywords:** heterocyclic amino acids, pyrazoles, piperidines, β-keto esters, enamines, hydrazines, building blocks

## Abstract

Series of methyl 3- and 5-(*N*-Boc-piperidinyl)-1*H*-pyrazole-4-carboxylates were developed and regioselectively synthesized as novel heterocyclic amino acids in their *N*-Boc protected ester form for achiral and chiral building blocks. In the first stage of the synthesis, piperidine-4-carboxylic and (*R*)- and (*S*)-piperidine-3-carboxylic acids were converted to the corresponding β-keto esters, which were then treated with *N*,*N*-dimethylformamide dimethyl acetal. The subsequent reaction of β-enamine diketones with various *N*-mono-substituted hydrazines afforded the target 5-(*N*-Boc-piperidinyl)-1*H*-pyrazole-4-carboxylates as major products, and tautomeric NH-pyrazoles prepared from hydrazine hydrate were further *N*-alkylated with alkyl halides to give 3-(*N*-Boc-piperidinyl)-1*H*-pyrazole-4-carboxylates. The structures of the novel heterocyclic compounds were confirmed by ^1^H-, ^13^C-, and ^15^N-NMR spectroscopy and HRMS investigation.

## 1. Introduction

Heterocyclic amino acids are becoming very important in modern drug discovery [1,2,3,4,5]. For instance, (*RS*)-piperidine-3-carboxylic acid (dl-nipecotic acid) is one of the most potent inhibitors of neuronal and glial γ-aminobutyric acid (GABA) uptake in vitro [6]. (*S*)-Pyrrolidinyl-2-carboxylic acid (l-proline) has been found to act as an agonist of the glycine receptor and of both the *N*-methyl-d-aspartate (NMDA) and non-NMDA ionotropic glutamate receptors [7].

Heterocyclic amino acids are also important scaffolds and building blocks for the preparation of heterocyclic systems, hybrids, and peptides [8,9,10,11]. For example, l-proline has been applied as a scaffold in the preparation of pyrrolizidine [12,13,14], pyrrolo[1,2-*c*][1,3]oxazole [15], pyrrolo[2,1-*c*][1,4]benzodiazepine [16], and benzo[*f*]pyrrolo[1,2-*a*][1,4]diazepine derivatives [17], while nipecotic and isonipecotic acids have given derivatives of heterospirocyclic 3-amino-2*H*-azirines [18,19]. Moreover, l-proline is a building block for *N*-(3-mercapto-2-d-methylpropanoyl)-l-proline, named captopril, which is used to regulate blood pressure [20]. d-Nipecotic acid, a building block for (*R*)-1-[4,4-bis-(3-methyl-2-thienyl)-3-butenyl]-3-piperidine carboxylic acid, named (*R*)-tiagabine, which amplifies neurotransmission of GABA, the predominant inhibitory neurotransmitter in the brain [21,22,23]. New derivatives of nipecotic acid, guvacin, and homo-β-proline are very potent and selective analogs of GABA uptake inhibitors [24,25,26].

The heterocyclic tripeptide Gly-Pro-Glu **I**, containing an l-proline residue, is a neuroprotective compound for the control of neurodegenerative processes such as Parkinson’s disease [27,28], while a proline peptidomimetic, faldaprevir **II**, was used as an experimental drug to treat hepatitis (Figure 1) [29,30,31]. The synthetically prepared derivative of the tripeptide (pyro)Glu-His-Pro(NH_2_) **III** has specific activity as a hypothalamic gland thyrotropin-releasing hormone [32]. Many aromatic heterocyclic amino acids, such as [5-amino-4-(*tert*-butoxycarbonyl)thiophen-2-yl]acetic acid, provide synthetic peptides, including enantiopure cyclic tetraamide **IV** [33], which are similar to compounds in marine plants that exhibit resistance to infection or antitumor effects [34].

Heterocyclic amino acids have been applied widely as building blocks for the preparation of DNA-encoded chemical libraries, including heterocyclic hybrid and peptide compounds [35,36,37,38,39]. In general, a DNA-encoded library of target component molecules should have a high degree of structural and functional diversity, taking into account diversity-oriented synthesis (DOS) [40]. For example, a highly specific and potent p38α kinase tripeptide-type inhibitor (VPC00628) **V** containing the residue of 3-amino-1-phenyl-1*H*-pyrazole-4-carboxylic acid has been identified directly from a multimillion-membered DNA-encoded molecule library that was prepared using high-fidelity yoctoReactor (yR) technology [41].

We recently reported an efficient protocol for synthesizing highly functionalized amino acid building blocks by combining pyrazole, indazole, and indole carboxylates with *N*-Boc-3-iodoazetidine [42]. Moreover, we synthesized 4-(*N*-Boc-cycloaminyl)-1,3-thiazole- and 4-(*N*-Boc-cycloaminyl)-1,3-selenazole-5-carboxylates as novel heterocyclic chiral amino acid-like derivatives [43,44]. Herein, we report the efficient synthesis of 3(5)-(*N*-Boc-piperidinyl)-1*H*-pyrazole-4-carboxylates as heterocyclic amino acid-like derivatives for novel achiral and chiral building blocks from piperidine-4-carboxylic and (*R*)- and (*S*)-piperidine-3-carboxylic acids.

## 2. Results and Discussion

Numerous methods for forming pyrazole ring systems have been developed. The most common synthetic method for the production of pyrazoles is the condensation of the corresponding hydrazine derivative, which acts as a double nitrogen nucleophile, with three carbon units containing compounds such as 1,3-dicarbonyl and 2,3-unsaturated carbonyl, or enamine [45,46,47]. Rosa et al. [48] developed a simple and efficient method for preparing both regioisomers of 4,5-substituted *N*-phenylpyrazoles from β-enamino diketones and phenylhydrazine, and the regiochemistry of the reaction was protic or aprotic solvent dependent. A patent [49] was obtained for the synthesis of 4-(piperidin-4-yl)-*N*-phenylpyrazole derivatives from β-enamino diketones with 4-fluoro- and 4-methoxyphenylhydrazines.

Our strategy for the synthesis of methyl 3(5)-(*N*-Boc-piperidinyl)-1*H*-pyrazole-4-carboxylates according to the enamine method is described in Scheme 1, Scheme 2 and Figure 2. The synthetic sequence started with preparing β-keto esters **2a**–**c** by treating *N*-Boc protected piperidine acids **1a**–**c** with 2,2-dimethyl-1,3-dioxane-4,6-dione (Meldrum’s acid) in the presence of 1-(3-dimethylaminopropyl)-3-ethylcarbodiimide hydrochloride (EDC·HCl) and 4-dimethylaminopyridine (DMAP), and further methanolysis of Meldrum’s acid adduct [50,51]. Compounds **2a**–**c** were treated with *N*,*N*-dimethylformamide dimethyl acetal (DMF·DMA) to obtain β-enamino diketones **3a**–**c** [49].

In the next step, we investigated the formation of 3(5)-substituted-1*H*-pyrazoles **5** and **6** via the key intermediates **4** and **4′** (Scheme 2). Optimization of the coupling reaction conditions was undertaken, choosing **3a** and phenylhydrazine as a model system (Table 1). An investigation of the reaction course and regioselectivity was carried out in various solvents, and the LC/MS and ^1^H-NMR spectral data of the crude reaction mixture of intermediate compound **4a** and products **5a**, **6a** were analyzed after 1 and 18 h (Table 1). EtOH was used as a polar protic (Table 1, entry 1), ACN as a polar aprotic (Table 1, entry 2), and CCl_4_ as a nonpolar solvent (Table 1, entry 3). As a result, the reaction in EtOH provided high regioselectivity (99.5%) and good yield (78%) of **5a** and just traces of its regioisomer **6a** (Table 1, entry 1). Similarly, the reaction in ACN resulted in **5a** as the main product (75%), and **6a** was obtained with a 3% yield (Table 1, entry 2). The poorest yield and regioselectivity were observed when the reaction mixture was stirred in CCl_4_. In this case, **5a** formed as a major product with 54% yield, and regioisomer **6a** was obtained with 9% yield (Table 1, entry 3). During optimization of the reaction conditions in different solvents, ^1^H-NMR analysis of the crude reaction mixture after 1 h also showed the formation of intermediate compound **4a**, which was successfully isolated for structure elucidation. The regioisomer **6a** formed as a minor isomer via intermediate **4′a** which resulted from the nucleophilic attack of a secondary amino group of phenylhydrazine on β-enamino diketone **3a**.

In the case of intermediate compound **4a**, the key information for structure elucidation was obtained from the ^15^N-NMR data. In the ^1^H-^15^N HMBC spectrum of **4a**, the ^15^N shift of δ −241.2 ppm was assigned to nitrogen N_a_, due to the correlation with the neighboring protons 2′(6′)-H (δ 6.81 ppm) from the phenyl moiety (Figure 3). The ^1^H-^15^N HSQC experiment indicated that proton N_a_-H (δ 6.24 ppm) had one-bond connectivity with the aforementioned nitrogen N_a_ at δ −241.2 ppm, while proton N_b_-H (δ 11.72 ppm) generated a cross peak with nitrogen N_b_ at δ −275.1 ppm. The formation of compound **4a** was also confirmed by a NOESY experiment, which exhibited NOEs between the 2′(6′)-H protons at δ 6.81 ppm and the enamine proton at δ 8.28 ppm. However, the configuration of the (2*E* or 2*Z*)-isomer of compound **4a** is not yet known.

Discrimination between regioisomeric compounds **5a** and **6a** was based on data from ^1^H-^13^C HMBC, ^1^H-^15^N HMBC, and ^1^H-^1^H NOESY experiments (Figure 3). The ^1^H-^15^N HMBC experiment of the major regioisomer **5a** revealed three-bond correlations between the piperidine 4′-H proton at δ 3.10 ppm and the phenyl group 2″(6″)-H protons at δ 7.34 ppm, with the pyrazole N-1 “pyrrole-like” nitrogen at δ –160.3 ppm [52,53]. The ^1^H-^1^H NOESY spectrum of **5a** exhibited NOEs between the phenyl group 2″(6″)-H protons and the 4′-H proton from the piperidine moiety.

The second regioisomer **6a** was easily identified by utilizing a similar approach. The minor regioisomer **6a** exhibited a strong three-bond connectivity between the piperidine proton 4′-H (δ 3.43 ppm) and the pyrazole N-2 “pyridine-like” nitrogen at δ –81.5 ppm, while the phenyl group protons 2″(6″)-H (δ 7.68 ppm) showed three-bond connectivity with the pyrazole N-1 “pyrrole-like” nitrogen at δ –165.7 ppm. Moreover, the pyrazole 5-H proton in the ^1^H-^13^C HMBC spectrum showed a three-bond connectivity with the phenyl group C-1″ carbon at δ 139.3 ppm. Finally, confirmation of these regiochemical assignments was obtained from the ^1^H-^1^H NOESY **6a** spectrum, showing only the NOEs between the phenyl group 2″(6″)-H protons and the pyrazole 5-H proton (δ 8.34 ppm).

The optimal conditions for the regioselective synthesis of methyl 5-(*N*-Boc-piperidinyl)-1*H*-pyrazole-4-carboxylate **5a** were applied to the synthesis of other pyrazoles to evaluate the scope of the methodology (Figure 2). β-Enamino diketone **3a** was coupled with different phenylhydrazines to give corresponding products **5b**–**h** with fair to good yields. No obvious effect of the phenylhydrazine substituent on the reaction yield was observed. A reaction of β-enamino diketone **3a** with methylhydrazine provided a corresponding *tert*-butyl 4-[4-(methoxycarbonyl)-1-methyl-1*H*-pyrazol-5-yl]piperidine-1-carboxylate **5i** with a 51% yield. To our delight, the reactions of chiral β-enamino diketones **3b**,**c** with phenyl-, (4-methylphenyl)- or [3-(trifluoromethyl)phenyl]hydrazines formed products **5j**–**o**, also with good yields. While analyzing the LC/MS and ^1^H-NMR spectral data of crude cyclization reaction mixtures, the formation of the regioisomeric **6b**–**o** was observed at trace amounts. The structure of compounds **5b**–**o** was determined by analogous NMR spectroscopy experiments as described above.

Next, having β-enamino diketone **3a**, we also performed a cyclocondensation reaction with hydrazine hydrate under the conditions described above, and the formation of tautomeric 3(5)-substituted NH-pyrazole **7** was established by NMR analysis (Scheme 3, Figure 4).

The prototropic tautomerism of NH-pyrazoles is well documented in many scientific studies, including with the use of multinuclear dynamic NMR spectroscopy [54,55,56]. In general, the annular tautomerism of 3(5)-1*H*-pyrazoles in solution under normal conditions is a very rapid process on the NMR time scale, and the determination of tautomeric ratios can usually be achieved only at low temperatures [57]. We carried out NMR studies of compound **7** at 25 °C in a diluted CDCl_3_ solution (Figure 4). The ^1^H-NMR spectrum of compound **7** revealed a narrow singlet of the pyrazole ring proton resonating at δ 7.96 [3(5)-H] and two singlets for methyl ester and Boc moiety protons in the area of δ 3.83 (OCH_3_) and 1.47 [C(CH_3_)_3_] ppm, respectively. The ^13^C-NMR spectrum provided important information; as expected, the characteristic signal of the pyrazole C-4 carbon at δ 110.1 ppm remained sharp, while the other two signals of pyrazole ring carbons 3(5)-C resonated at δ 138.7 and 153.6 ppm and appeared broadened. It is known that the broadening of NMR spectral lines very often reflects dynamic structural transformations of molecules in solution [58]. Therefore, the observed broadness of relevant C-3 and C-5 pyrazole carbon signals is due to the coalescence of individual signals to average signals, indicating tautomeric equilibrium of **7** (**7a** and **7b**). In addition, the pyrazole NH proton (δ 11.52 ppm) exhibited NOEs not only with the pyrazole ring proton at 7.96 ppm but also with the 3′-H piperidine protons at 1.70 ppm, which is only possible in the case of annular tautomerism **7**. It was not possible to obtain relevant information for the nitrogen atoms of the pyrazole ring N-1 and N-2 from the ^15^N-NMR spectral data since ^1^H-^15^N HSQC and HMBC experiments showed no direct or long-range correlations with appropriate protons.

Tautomeric compound **7** was alkylated with alkyl iodides (Scheme 3). It is known that *N*-alkylation of asymmetrically ring-substituted 1*H*-pyrazoles generally results in the formation of a mixture of regioisomeric *N*-substituted products [59]. Treatment of compound **7** with methyl iodide in the presence of KOH in DMF gave an inseparable mixture of regioisomers **5i** and **6i** in a ratio of about 1:5 and a total yield of 74%. However, alkylation of 1*H*-pyrazole-4-carboxylate **7** with ethyl iodide under analogous conditions afforded compound **8** as the sole product with a good 87% yield.

Discrimination of regioisomeric compounds **5i** and **6i** were based on ^1^H-^13^C HMBC, ^1^H-^15^N HMBC, and ^1^H-^1^H NOESY spectral data (Figure 4). In the ^1^H-^15^N HMBC spectra of minor regioisomer **5i**, a ^15^N shift of δ −178.3 ppm was assigned to the “pyrrole-like” nitrogen N-1 due to the correlation of this signal with a piperidine ring proton 4′-H (δ 3.54 ppm). The ^1^H-^13^C HMBC experiment exhibited a three-bond correlation of the 1-CH_3_ protons with a pyrazole quaternary carbon C-5 at δ 148.7 ppm. Moreover, the ^1^H-^1^H NOESY spectrum of **5i** exhibited NOEs between the methyl group protons (1-CH_3_) at 3.92 ppm and the piperidine proton 4′-H at δ 3.54 ppm. In the ^1^H-^15^N HMBC spectra of the major regioisomer **6i**, an appropriate correlation between the piperidine ring proton 4′-H (δ 3.36 ppm) and the “pyridine-like” pyrazole N-2 nitrogen which resonated at δ −77.3 ppm could be observed. The ^1^H-^13^C HMBC spectral data of compound **6i** provided a strong three-bond correlation of 1-CH_3_ protons with pyrazole protonated carbon C-5 at δ 134.6 ppm. Finally, the regiochemistry of compound **6i** was confirmed by a NOESY experiment, which exhibited NOEs between the 1-CH_3_ protons and pyrazole proton 5-H (δ 7.78 ppm). The structure of compound **8** was determined by analogous NMR spectroscopy experiments as described above.

After the successful synthesis of 3(5)-(*N*-Boc-piperidinyl)-1*H*-pyrazole-4-carboxylates, we further prepared several pyrazole carboxylic acids (Scheme 4). In particular, achiral pyrazole-4-carboxylic acid **9a** was prepared from the corresponding ester **5a** under the basic conditions (2N NaOH, methanol, reflux). The same hydrolysis conditions were applied to the production of chiral pyrazole-4-carboxylic acids (*R*)-**9b** and (*S*)-**9c** from esters **5j** and **5k**, respectively.

Pyrazole carboxylic acid amides, including anilides, have been known to play an important role in agrochemical research as fungicides [60,61]. Pyrazole-4-carboxylic acids **9a**–**c** were used to obtain new anilide compounds (Scheme 4). First, **9a** reacted with aniline in the presence of EDC·HCl, DMAP, and dichloromethane to give pyrazole anilide **10a**. Moreover, chiral pyrazole anilide (*R*)-**10b** (100% ee) was obtained from carboxylic acid **9b**, while the corresponding chiral anilide (*S*)-**10c** (96% ee) was synthesized from carboxylic acid **9c**. The enantiomeric purity of prepared anilides **10b**,**c** was evaluated by chiral HPLC analysis. As an example, HPLC analysis of enantiomeric samples of anilides **10b**,**c** is shown in Figure 5.

## 3. Materials and Methods

### 3.1. General Information

All starting materials were purchased from commercial suppliers and were used as received. Flash column chromatography was performed on Silica Gel 60 Å (230–400 µm, Merck KGaA, Darmstadt, Germany). Thin-layer chromatography was carried out on Silica Gel plates (Merck Kieselgel 60 F_254_) and visualized by UV light (254 nm). Melting points were determined on a Büchi M-565 melting point apparatus and were uncorrected. The IR spectra were recorded on a Bruker Vertex 70v FT-IR spectrometer (Bruker Optik GmbH, Ettlingen, Germany) using neat samples and are reported in the frequency of absorption (cm^–1^). Mass spectra were obtained on a Shimadzu LCMS-2020 (ESI^+^) spectrometer (Shimadzu Corporation, Kyoto, Japan). High-resolution mass spectra were measured on Bruker MicrOTOF-Q III (ESI^+^) apparatus (Bruker Daltonik GmbH, Bremen, Germany). Optical rotation data were recorded on a UniPol L SCHMIDT+HAENSCH polarimeter (concentration of compound (g/100 mL) was included in calculations automatically (Windaus-Labortechnik GmbH & Co. KG, Clausthal-Zellerfeld, Germany). HPLC analysis was carried out on Shimadzu LC-2030C apparatus with CHIRAL ART Amylose-SA (100 × 4.6 mm I.D.; S-3 µm; chiral selector amylose tris(3,5-dimethylphenylcarbamate); YMC, Shimadzu USA Manufacturing, Inc., Canby, OR, USA). The ^1^H-, ^13^C-, and ^15^N-NMR spectra were recorded in CDCl_3_ solutions at 25 °C on a Bruker Avance III 700 (700 MHz for ^1^H, 176 MHz for ^13^C, 71 MHz for ^15^N, Bruker BioSpin AG, Fallanden, Switzerland) spectrometer equipped with a 5 mm TCI ^1^H-^13^C/^15^N/D z-gradient cryoprobe, and a Bruker Avance III 400 (400 MHz for ^1^H, 101 MHz for ^13^C, 40 MHz for ^15^N, (Bruker BioSpin AG) spectrometer using a 5 mm directly detecting BBO probe. The chemical shifts (δ) expressed in ppm, were relative to tetramethylsilane (TMS). The ^15^N-NMR spectra were referenced to neat, external nitromethane (coaxial capillary). Full and unambiguous assignment of the ^1^H-, ^13^C- and ^15^N-NMR resonances was achieved using a combination of standard NMR spectroscopic techniques [62] such as DEPT, COSY, gs-HSQC, gs-HMBC, NOESY and 1,1-ADEQUATE experiments [63]. ^1^H-, ^13^C-, and ^1^H-^15^N HMBC NMR spectra, and HRMS data of all new compounds are provided in Appendix A as Appendix A.

### 3.2. Synthesis of tert-Butyl 3- and 4-[(2)-3-(Dimethylamino)-2-(methoxycarbonyl)prop-2-enoyl]piperidine-1-carboxylates (***3a***–***c***)

To a solution of the corresponding l-(*tert*-butoxycarbonyl)piperidinecarboxylic acid (**1a**–**c**) (4 g, 17.4 mmol) in DCM (24 mL) cooled to 0 °C temperature Meldrum’s acid (2.77 g, 19.2 mmol) was added followed by DMAP (4.26 g, 34.9 mmol). Then EDC⋅HCl (3.68 g, 19.2 mmol) was added in portions over 10 min. The reaction mixture was gradually warmed to r.t. and stirred for 16 h. The reaction solution was diluted with DCM (10 mL), washed with 1 M KHSO_4_ (2 × 15 mL) and brine (20 mL). The organic layer was dried with anhydrous sodium sulfate, filtered, and concentrated under reduced pressure. Then the residue was dissolved in MeOH (20 mL) and left under reflux for 5 h. The solvent was evaporated in vacuo. A solution of crude β-keto ester (**2a**–**c**) (4.7 g, 16.4 mmol) and *N*,*N*-dimethylformamide dimethyl acetal (4.4 mL, 32.8 mmol) in dioxane (24 mL) was stirred at 100 °C. After 5 h the solvent was removed under reduced pressure. Crude compounds **3a**–**c** were carried forward without any further purification.

### 3.3. Synthesis Procedure for the Preparation of Compounds ***4a***, ***5a***, and ***6a***

Method I. Compound **3a** (500 mg, 1.5 mmol) was dissolved in EtOH (15 mL) and treated with phenylhydrazine (160 mg, 1.5 mmol). The reaction mixture was stirred at r.t. for 18 h. After removal of the solvent in vacuo, the residue was purified by flash column chromatography (SiO_2_, eluent: acetone/*n*-hexane, 1:7, *v*/*v*) to provide compound **5a** (441 mg, 78%).

Method II. The reaction of compound **3a** (500 mg, 1.5 mmol) with phenylhydrazine (160 mg, 1.5 mmol) in ACN (15 mL), was carried out and purified as described in Method I and afforded compounds **5a** (424 mg, 75%) and **6a** (17 mg, 3%).

Method III. The reaction of compound **3a** (500 mg, 1.5 mmol) with phenylhydrazine (160 mg, 1.5 mmol) in CCl_4_ (15 mL) was carried out as described in Method I, and the resulted residue was purified by gradient flash chromatography on silica gel (acetone/*n*-hexane, 1:15→1:7, *v*/*v*) to yield compounds **4a** (79 mg, 14%), **5a** (305 mg, 54%) and **6a** (51 mg, 9%).

#### 3.3.1. *tert*-Butyl 4-[(2*E*(*Z*))-2-(methoxycarbonyl)-3-(2-phenylhydrazinyl)prop-2-enoyl]piperidine-1-carboxylate (**4a**)

Yellowish oil. ^1^H-NMR (700 MHz, CDCl_3_): δ 1.46 (s, 9H, C(CH_3_)_3_), 1.53–1.60 (m, 2H, Pip 3,5-H), 1.75–1.83 (m, 2H, Pip 3,5-H), 2.76–2.87 (m, 2H, Pip 2,6-H), 3.70 (tt, *J* = 11.7 Hz, 3.5 Hz, 1H, Pip 4-H), 3.73 (s, 3H, OCH_3_), 4.06–4.26 (m, 2H, Pip 2,6-H), 6.24 (s, 1H, N_a_H), 6.81 (d, *J* = 8.1 Hz, 2H, Ph 2‘,6‘-H), 6.99 (t, *J* = 7.4 Hz, 1H, Ph 4‘-H), 7.28 (t, *J* = 7.8 Hz, 2H, Ph 3‘,5‘-H), 8.28 (d, *J* = 10.8 Hz, 1H, 3*E(Z)*-H), 11.76 (d, *J* = 10.8 Hz, 1H, N_b_H). ^13^C-NMR (176 MHz, CDCl_3_): δ 28.5 (2 × CH_2_, Pip 3,5-C and C(*C*H_3_)_3_), 43.8 (2 × CH_2_, Pip 2,6-C), 45.5 (Pip 4-C), 51.1 (OCH_3_), 79.3 (*C*(CH_3_)_3_), 99.2 (2*E(Z)*-C), 113.6 (2 × CH, Ph 2‘,6‘-C), 122.4 (Ph 4‘-C), 129.5 (2 × CH, Ph 3‘,5‘-C), 146.3 (Ph 1‘-C), 154.8 (*C*OOC(CH_3_)_3_), 162.4 (3*E(Z)*-C), 166.6 (*C*OOCH_3_), 203.5 (C=O). ^15^N-NMR (71 MHz, CDCl_3_): δ −275.1 (N_b_H), −241.2 (N_a_H). IR (FT-IR, ν_max_, cm^−1^): 3438(N-H), 2928, 1717 (C=O), 1690 (C=O), 1242, 767. MS *m*/*z* (%): 402 ([M − H]^−^, 95%). HRMS (ESI^+^) for C_21_H_29_N_3_NaO_5_ ([M + Na]^+^) calcd 426.1999, found 426.2001.

#### 3.3.2. *tert*-Butyl 4-[4-(methoxycarbonyl)-1-phenyl-1*H*-pyrazol-5-yl]piperidine-1-carboxylate (**5a**)

Yellowish crystals, mp 151–153 °C. ^1^H-NMR (700 MHz, CDCl_3_): δ 1.46 (s, 9H, C(CH_3_)_3_), 1.53–1.61 (m, 2H, Pip 3,5-H), 2.28 (qd, *J* = 12.7 Hz, 4.5 Hz, 2H, Pip 3,5-H), 2.49–2.69 (m, 2H, Pip 2,6-H), 3.10 (tt, *J* = 12.4 Hz, 3.6 Hz, 1H, Pip 4-H), 3.84 (s, 3H, OCH_3_), 4.02–4.27 (m, 2H, Pip 2,6-H), 7.31–7.37 (m, 2H, Ph 2,6-H), 7.47–7.54 (m, 3H, Ph 3,4,5-H), 8.03 (s, 1H, Pyr 3-H). ^13^C-NMR (176 MHz, CDCl_3_): δ 28.4 (C(*C*H_3_)_3_), 28.6 (2 × CH_2_, Pip 3,5-C), 35.1 (Pip 4-C), 44.1 (2 × CH_2_, Pip 2,6-C), 51.2 (OCH_3_), 79.4 (*C*(CH_3_)_3_), 111.7 (Pyr 4-C), 126.6 (2 × CH, Ph 2,6-C), 129.3 (2 × CH, Ph 3,5-C), 129.4 (Ph 4-C), 139.2 (Ph 1-C), 142.8 (Pyr 3-C), 149.8 (Pyr 5-C), 154.7 (*C*OOC(CH_3_)_3_), 163.5 (*C*OOCH_3_). ^15^N-NMR (71 MHz, CDCl_3_): δ −294.5 (N-Boc), −160.3 (Pyr N-1), −76.0 (Pyr N-2). IR (FT-IR, ν_max_, cm^−1^): 2979, 1712 (C=O), 1674 (C=O), 1255, 765. MS *m*/*z* (%): 386 ([M + H]^+^, 99%). HRMS (ESI^+^) for C_21_H_27_N_3_NaO_4_ ([M + Na]^+^) calcd 408.1894, found 408.1894.

#### 3.3.3. *tert*-Butyl 4-[4-(methoxycarbonyl)-1-phenyl-1*H*-pyrazol-3-yl]piperidine-1-carboxylate (**6a**)

Brownish crystals, mp 134–136 °C. ^1^H-NMR (700 MHz, CDCl_3_): δ 1.47 (s, 9H, C(CH_3_)_3_), 1.83 (qd, *J* = 12.2 Hz, 4.2 Hz, 2H, Pip 3,5-H), 1.93–2.01 (m, 2H, Pip 3,5-H), 2.83–2.97 (m, 2H, Pip 2,6-H), 3.43 (tt, *J* = 11.6 Hz, 3.7 Hz, 1H, Pip 4-H), 3.85 (s, 3H, OCH_3_), 4.12–4.29 (m, 2H, Pip 2,6-H), 7.30–7.35 (m, 1H, Ph 4-H), 7.43–7.49 (m, 2H, Ph 3,5-H), 7.65–7.70 (m, 2H, Ph 2,6 -H), 8.34 (s, 1H, Pyr 5-H). ^13^C-NMR (176 MHz, CDCl_3_): δ 28.6 (C(*C*H_3_)_3_), 31.0 (2 × CH_2_, Pip 3,5-C), 35.0 (Pip 4-C), 44.0 (2 × CH_2_, Pip 2,6-C), 51.4 (OCH_3_), 79.4 (*C*(CH_3_)_3_), 112.8 (Pyr 4-C), 119.5 (2 × CH, Ph 2,6-C), 127.3 (Ph 4-C), 129.7 (2 × CH, Ph 3,5-C), 131.2 (Pyr 5-C), 139.3 (Ph 1-C), 155.0 (*C*OOC(CH_3_)_3_), 159.0 (Pyr 3-C), 163.8 (*C*OOCH_3_**).**
^15^N-NMR (71 MHz, CDCl_3_): δ −292.6 (N-Boc), −165.7 (Pyr N-1), −81.5 (Pyr N-2). IR (FT-IR, ν_max_, cm^−1^): 2949, 1708 (C=O), 1692 (C=O), 1537, 753. MS *m*/*z* (%): 386 ([M + H]^+^, 95%). HRMS (ESI^+^) for C_21_H_27_N_3_NaO_4_ ([M + Na]^+^) calcd 408.1894, found 408.1894.

### 3.4. Synthesis of tert-Butyl 4-[4-(methoxycarbonyl)-1H-pyrazol-5-yl]piperidine-1-carboxylates (***5b***–***o***)

Compounds **5b**–**o** were obtained from β-enamino diketones **3a**–**c** (500 mg, 1.5 mmol) and appropriate hydrazines (1.5 mmol) in EtOH (15 mL) by the procedure which was used for the preparation of compound **5a**
*(*Method I*)*.

#### 3.4.1. *tert*-Butyl 4-[4-(methoxycarbonyl)-1-(4-methylphenyl)-1*H*-pyrazol-5-yl]piperidine-1-carboxylate (**5b**)

Compound **3a** was coupled with *p*-tolylhydrazine hydrochloride. The obtained residue was purified by column chromatography (SiO_2_, eluent: acetone/*n*-hexane, 1:7, *v*/*v*) to provide compound **5b** as yellowish crystals. Yield 411 mg (70%), mp 133–135 °C. ^1^H-NMR (400 MHz, CDCl_3_): δ 1.44 (s, 9H, C(CH_3_)_3_), 1.49–1.61 (m, 2H, Pip 3,5-H), 2.23 (qd, *J* = 12.6 Hz, 4.3 Hz, 2H, Pip 3,5-H), 2.44 (s, 3H, CH_3_), 2.49–2.72 (m, 2H, Pip 2,6-H), 3.10 (tt, *J* = 12.4 Hz, 3.6 Hz, 1H, Pip 4-H), 3.83 (s, 3H, OCH_3_), 4.00–4.29 (m, 2H, Pip 2,6-H), 7.17–7.23 (m, 2H, Ph 2,6-H), 7.27–7.32 (m, 2H, Ph 3,5-H), 8.00 (s, 1H, Pyr 3-H). ^13^C-NMR (101 MHz, CDCl_3_): δ 21.4 (CH_3_), 28.6 (C(*C*H_3_)_3_), 28.9 (2 × CH_2_, Pip 3,5-C), 35.3 (Pip 4-C), 44.4 (2 × CH_2_, Pip 2,6-C), 51.4 (OCH_3_), 79.6 (*C*(CH_3_)_3_), 111.8 (Pyr 4-C), 126.5 (2 × CH, Ph 2,6-C), 130.0 (2 × CH, Ph 3,5-C), 137.0 (Ph 1-C), 139.7 (Ph 4-C), 142.9 (Pyr 3-C), 150.1 (Pyr 5-C), 155.0 (*C*OOC(CH_3_)_3_), 163.8 (*C*OOCH_3_). ^15^N-NMR (41 MHz, CDCl_3_): δ −294.5 (N-Boc), −160.6 (Pyr N-1), −75.8 (Pyr N-2). IR (FT-IR, ν_max_, cm^−1^): 2980, 1703 (C=O), 1688 (C=O), 1255, 779. MS *m*/*z* (%): 400 ([M + H]^+^, 99%). HRMS (ESI^+^) for C_22_H_29_N_3_NaO_4_ ([M + Na]^+^) calcd 422.2050, found 422.2051.

#### 3.4.2. *tert*-Butyl 4-[4-(methoxycarbonyl)-1-(3-methylphenyl)-1*H*-pyrazol-5-yl]piperidine-1-carboxylate (**5c**)

Compound **3a** was coupled with *m*-tolylhydrazine hydrochloride. The obtained residue was purified by column chromatography (SiO_2_, eluent: acetone/*n*-hexane, 1:9, *v/v*) to provide compound **5c** as white crystals. Yield 310 mg (53%), mp 123–124 °C. ^1^H-NMR (400 MHz, CDCl_3_): δ 1.45 (s, 9H, C(CH_3_)_3_), 1.51–1.60 (m, 2H, Pip 3,5-H), 2.26 (qd, *J* = 12.7 Hz, 4.4 Hz, 2H, Pip 3,5-H), 2.42 (s, 3H, CH_3_), 2.50–2.71 (m, 2H, Pip 2,6-H), 3.08 (tt, *J* = 12.4 Hz, 3.6 Hz, 1H, Pip 4-H), 3.83 (s, 3H, OCH_3_), 4.04–4.26 (m, 2H, Pip 2,6-H), 7.10 (d, *J* = 7.9 Hz, 1H, Ph 6-H), 7.16 (s, 1H, Ph 2-H), 7.30 (d, *J* = 8.0 Hz, 1H, Ph 4-H), 7.37 (t, *J* = 7.7 Hz, 1H, Ph 5-H), 8.01 (s, 1H, Pyr 3-H). ^13^CNMR (101 MHz, CDCl_3_): δ 21.4 (CH_3_), 28.6 (C(*C*H_3_)_3_), 28.8 (2 × CH_2_, Pip 3,5-C), 35.3 (Pip 4-C), 44.3 (2 × CH_2_, Pip 2,6-C), 51.4 (O*C*H_3_), 79.6 (*C*(CH_3_)_3_), 111.8 (Pyr 4-C), 123.6 (Ph 6-C), 127.4 (Ph 2-C), 129.1 (Ph 5-C), 130.3 (Ph 4-C), 139.4 (Ph 1-C), 139.8 (Ph 3-C), 142.9 (Pyr 3-C), 150.0 (Pyr 5-C), 155.0 (*C*OOC(CH_3_)_3_), 163.8 (*C*OOCH_3_). ^15^N-NMR (41 MHz, CDCl_3_): δ −159.9 (Pyr N-1), −76.2 (Pyr N-2). IR (FT-IR, ν_max_, cm^−1^): 2979, 1703 (C=O), 1688 (C=O), 1243, 779. MS *m*/*z* (%): 400 ([M + H]^+^, 100%). HRMS (ESI^+^) for C_22_H_29_N_3_NaO_4_ ([M + Na]^+^) calcd 422.2050, found 422.2050.

#### 3.4.3. *tert*-Butyl 4-[1-(3-fluorophenyl)-4-(methoxycarbonyl)-1*H*-pyrazol-5-yl]piperidine-1-carboxylate (**5d**)

Compound **3a** was coupled with (3-fluorophenyl)hydrazine hydrochloride. The obtained residue was purified by column chromatography (SiO_2_, eluent: acetone/*n*-hexane, 1:8, *v*/*v*) to provide compound **5d** as yellowish crystals. Yield 433 mg (73%), mp 134–136 °C. ^1^H-NMR (400 MHz, CDCl_3_): δ 1.45 (s, 9H, C(CH_3_)_3_), 1.51–1.64 (m, 2H, Pip 3,5-H), 2.27 (qd, *J* = 12.7 Hz, 4.4 Hz, 2H, Pip 3,5-H), 2.49–2.72 (m, 2H, Pip 2,6-H), 3.08 (tt, *J* = 12.3 Hz, 3.6 Hz, 1H, Pip 4-H), 3.83 (s, 3H, OCH_3_), 3.96–4.32 (m, 2H, Pip 2,6-H), 7.05–7.17 (m, 2H, Ph 2,6-H), 7.19–7.25 (m, 1H, Ph 4-H), 7.43–7.53 (m, 1H, Ph 5-H), 8.02 (s, 1H, Pyr 3-H). ^13^C-NMR (101 MHz, CDCl_3_): δ 28.6 (C(*C*H_3_)_3_), 28.8 (2 × CH_2_, Pip 3,5-C), 35.3 (Pip 4-C), 44.3 (2 × CH_2_, Pip 2,6-C), 51.5 (OCH_3_), 79.7 (*C*(CH_3_)_3_), 112.3 (Pyr 4-C), 114.6 (d, *J* = 23.8 Hz, Ph 2-C), 116.8 (d, *J* = 20.9 Hz, Ph 4-C), 122.5 (d, *J* = 3.3 Hz, Ph 6-C), 130.7 (d, *J* = 9.0 Hz, Ph 5-C), 140.7 (d, *J* = 9.7 Hz, Ph 1-C), 143.3 (Pyr 3-C), 150.1 (Pyr 5-C), 154.9 (*C*OOC(CH_3_)_3_), 162.8 (d, *J* = 249.7 Hz, Ph 3-C), 163.5 (*C*OOCH_3_). ^15^N-NMR (41 MHz, CDCl_3_): δ −161.2 (Pyr N-1), −74.5 (Pyr N-2). IR (FT-IR, ν_max_, cm^−1^): 2980, 1711 (C=O), 1674 (C=O), 1243, 867. MS *m/z* (%): 404 ([M + H]^+^, 99%). HRMS (ESI^+^) for C_21_H_26_FN_3_NaO_4_ ([M + Na]^+^) calcd 426.1800, found 426.1799.

#### 3.4.4. *tert*-Butyl 4-[1-(2-fluorophenyl)-4-(methoxycarbonyl)-1*H*-pyrazol-5-yl]piperidine-1-carboxylate (**5e**)

Compound **3a** was coupled with (2-fluorophenyl)hydrazine hydrochloride. The obtained residue was purified by column chromatography (SiO_2_, eluent: acetone/*n*-hexane, 1:8, *v*/*v*) to provide compound **5e** as yellowish crystals. Yield 367 mg (62%), mp 114–116 °C. ^1^H-NMR (400 MHz, CDCl_3_): δ 1.43 (s, 9H, C(CH_3_)_3_), 1.48–1.58 (m, 1H, Pip 3-H), 1.60–1.76 (m, 1H, Pip 5-H), 2.03–2.30 (m, 2H, Pip 3,5-H), 2.44–2.71 (m, 2H, Pip 2,6-H), 2.88–3.05 (m, 1H, Pip 4-H), 3.83 (s, 3H, OCH_3_), 3.95–4.29 (m, 2H, Pip 2,6-H), 7.22–7.32 (m, 2H, Ph 3,6-H), 7.34–7.42 (m, 1H, Ph 5-H), 7.47–7.55 (m, 1H, Ph 4-H), 8.06 (s, 1H, Pyr 3-H). ^13^C-NMR (101 MHz, CDCl_3_): δ 28.6 (C(*C*H_3_)_3_ and 2 × CH_2_, Pip 3,5-C), 35.6 (Pip 4-C), 44.1 (2 × CH_2_, Pip 2,6-C), 51.4 (OCH_3_), 79.6 (*C*(CH_3_)_3_), 111.8 (Pyr 4-C), 116.9 (d, *J* = 19.6 Hz, Ph 3-C), 125.0 (d, *J* = 4.0 Hz, Ph 6-C), 127.4 (d, *J* = 12.6 Hz, Ph 1-C), 129.7 (Ph 5-C), 131.8 (d, *J* = 7.7 Hz, Ph 4-C), 143.7 (Pyr 3-C), 151.4 (Pyr 5-C), 154.9 (*C*OOC(CH_3_)_3_), 157.5 (d, *J* = 252.5 Hz, Ph 2-C), 163.6 (*C*OOCH_3_). ^15^N-NMR (41 MHz, CDCl_3_): δ −292.8 (N-Boc), −172.7 (Pyr N-1), −73.8 (Pyr N-2). IR (FT-IR, ν_max_, cm^−1^): 2980, 1716 (C=O), 1682 (C=O), 1275, 770. MS *m*/*z* (%): 404 ([M + H]^+^, 96%). HRMS (ESI^+^) for C_21_H_26_FN_3_NaO_4_ ([M + Na]^+^) calcd 426.1800, found 426.1800.

#### 3.4.5. *tert*-Butyl 4-[4-(methoxycarbonyl)-1-(4-methoxyphenyl)-1*H*-pyrazol-5-yl]piperidine-1-carboxylate (**5f**)

Compound **3a** was coupled with (4-methoxyphenyl)hydrazine hydrochloride. The obtained residue was purified by column chromatography (SiO_2_, eluent: acetone/*n*-hexane, 1:8, *v/v*) to provide compound **5f** as orange crystals. Yield 366 mg (60%), mp 151–153 °C. ^1^H-NMR (400 MHz, CDCl_3_): δ 1.44 (s, 9H, C(CH_3_)_3_), 1.48–1.59 (m, 2H, Pip 3,5-H), 2.20 (qd, *J* = 12.7 Hz, 5.1 Hz, 2H, Pip 3,5-H), 2.49–2.73 (m, 2H, Pip 2,6-H), 3.09 (tt, *J* = 12.4 Hz, 3.6 Hz, 1H, Pip 4-H), 3.83 (s, 3H, COOCH_3_), 3.87 (s, 3H, OCH_3_), 3.99–4.30 (m, 2H, Pip 2,6-H), 6.98 (d, *J* = 8.8 Hz, 2H, Ph 3,5-H), 7.23 (d, *J* = 8.8 Hz, 2H, Ph 2,6-H), 7.99 (s, 1H, Pyr 3-H). ^13^C-NMR (101 MHz, CDCl_3_): δ 28.6 (C(*C*H_3_)_3_), 28.9 (2 × CH_2_, Pip 3,5-C), 35.3 (Pip 4-C), 44.2 (2 × CH_2_, Pip 2,6-C), 51.4 (COO*C*H_3_), 55.7 (OCH_3_), 79.6 (*C*(CH_3_)_3_), 111.6 (Pyr 4-C), 114.5 (2 × CH, Ph 3,5-C), 128.0 (2 × CH, Ph 2,6-C), 132.4 (Ph 1-C), 142.7 (Pyr 3-C), 150.2 (Pyr 5-C), 154.9 (*C*OOC(CH_3_)_3_), 160.3 (Ph 4-C), 163.8 (*C*OOCH_3_). ^15^N-NMR (41 MHz, CDCl_3_): δ −161.4 (Pyr N-1), −75.4 (Pyr N-2). IR (FT-IR, ν_max_, cm^−1^): 2981, 1715 (C=O), 1682 (C=O), 1245, 780. MS *m*/*z* (%): 416 ([M+H]^+^, 100%). HRMS (ESI^+^) for C_22_H_29_N_3_NaO_5_ ([M+Na]^+^) calcd 438.1999, found 438.2000.

#### 3.4.6. *tert*-Butyl 4-[4-(methoxycarbonyl)-1-(3-methoxyphenyl)-1*H*-pyrazol-5-yl]piperidine-1-carboxylate (**5g**)

Compound **3a** was coupled with (3-methoxyphenyl)hydrazine hydrochloride. The obtained residue was purified by column chromatography (SiO_2_, eluent: acetone/*n*-hexane, 1:9, *v*/*v*) to provide compound **5g** as yellowish crystals. Yield 330 mg (54%), mp 69–71 °C. ^1^H-NMR (400 MHz, CDCl_3_): δ 1.45 (s, 9H, C(CH_3_)_3_), 1.50–1.61 (m, 2H, Pip 3,5-H), 2.27 (qd, *J* = 12.7 Hz, 4.3 Hz, 2H, Pip 3,5-H), 2.51–2.71 (m, 2H, Pip 2,6-H), 3.11 (tt, *J* = 12.4 Hz, 3.7 Hz, 1H, Pip 4-H), 3.83 (s, 6H, OCH_3_ and COOCH_3_), 3.99–4.29 (m, 2H, Pip 2,6-H), 6.85–6.93 (m, 2H, Ph 2,4-H), 7.00–7.06 (m, 1H, Ph 6-H), 7.35–7.43 (m, 1H, Ph 5-H), 8.01 (s, 1H, Pyr 3-H). ^13^C-NMR (101 MHz, CDCl_3_): δ 28.6 (C(*C*H_3_)_3_), 28.8 (2 × CH_2_, Pip 3,5-C), 35.3 (Pip 4-C), 44.3 (2 × CH_2_, Pip 2,6-C), 51.4 (COO*C*H_3_), 55.7 (OCH_3_), 79.6 (*C*(CH_3_)_3_), 112.0 (Pyr 4-C), 112.5 (Ph 2-C), 115.5 (Ph 4-C), 118.8 (Ph 6-C), 130.1 (Ph 5-C), 140.4 (Ph 1-C), 143.0 (Pyr 3-C), 150.0 (Pyr 5-C), 155.0 (*C*OOC(CH_3_)_3_), 160.4 (Ph 3-C), 163.7 (*C*OOCH_3_). IR (FT-IR, ν_max_, cm^−1^): 2979, 1702 (C=O), 1686 (C=O), 1109, 778. MS *m*/*z* (%): 400 ([M + H]^+^, 100%). HRMS (ESI^+^) for C_22_H_29_N_3_NaO_5_ ([M + Na]^+^) calcd 438.1999, found 438.2000.

#### 3.4.7. *tert*-Butyl 4-{4-(methoxycarbonyl)-1-[3-(trifluoromethyl)phenyl]-1*H*-pyrazol-5-yl}piperidine-1-carboxylate (**5h**)

Compound **3a** was coupled with [3-(trifluoromethyl)phenyl]hydrazine hydrochloride. The obtained residue was purified by column chromatography (SiO_2_, eluent: acetone/*n*-hexane, 1:5, *v*/*v*) to provide compound **5h** as yellowish crystals. Yield 506 mg (76%), mp 113–115 °C. ^1^H-NMR (700 MHz, CDCl_3_): δ 1.46 (s, 9H, C(CH_3_)_3_), 1.55–1.62 (m, 2H, Pip 3,5-H), 2.24–2.37 (m, 2H, Pip 3,5-H), 2.50–2.70 (m, 2H, Pip 2,6-H), 3.05 (tt, *J* = 12.3 Hz, 3.6 Hz, 1H, Pip 4-H), 3.86 (s, 3H, OCH_3_), 4.06–4.29 (m, 2H, Pip 2,6-H), 7.54–7.57 (m, 1H, Ph 6-H), 7.65 (br s, 1H, Ph 2-H), 7.68 (t, *J* = 7.9 Hz, 1H, Ph 5-H), 7.79 (br s, 1H, Ph 4-H), 8.06 (s, 1H, Pyr 3-H). ^13^C-NMR (176 MHz, CDCl_3_): δ 28.4 (C(*C*H_3_)_3_), 28.8 (2 × CH_2_, Pip 3,5-C), 35.4 (Pip 4-C), 44.1 (2 × CH_2_, Pip 2,6-C), 51.4 (OCH_3_), 79.6 (*C*(CH_3_)_3_), 112.5 (Pyr 4-C), 123.3 (q, *J* = 271.5 Hz, CF_3_), 123.7 (q, *J* = 3.7 Hz, Ph 2-C), 126.2 (q, *J* = 3.7 Hz, Ph 4-C), 129.7 (Ph 5-C), 130.1 (Ph 6-C), 132.1 (q, *J* = 33.3 Hz, Ph 3-C), 139.8 (Ph 1-C), 143.5 (Pyr 3-C), 150.1 (Pyr 5-C), 154.8 (*C*OOC(CH_3_)_3_), 163.3 (*C*OOCH_3_). ^15^N-NMR (71 MHz, CDCl_3_): δ −294.7 (N-Boc), −163.3 (Pyr N-1), −76.0 (Pyr N-2). IR (FT-IR, ν_max_, cm^−1^): 2980, 1714 (C=O), 1686 (C=O), 1169, 1062. MS *m*/*z* (%): 354 ([M-Boc + H]^+^), 454 ([M + H]^+^), 99%. HRMS (ESI^+^) for C_22_H_26_F_3_N_3_NaO_4_ ([M + Na]^+^) calcd 476.1768, found 476.1768.

#### 3.4.8. *tert*-Butyl 4-[4-(methoxycarbonyl)-1-methyl-1*H*-pyrazol-5-yl]piperidine-1-carboxylate (**5i**)

Compound **3a** was coupled with methylhydrazine. The obtained residue was purified by column chromatography (SiO_2_, eluent: acetone/*n*-hexane, 1:3, *v/v*) to provide compound **5i** as white crystals. Yield 242 mg (51%), mp 147–149 °C. ^1^H NMR (400 MHz, CDCl_3_): δ 1.48 (s, 9H, C(CH_3_)_3_), 1.59–1.70 (m, 2H, Pip 3,5-H), 2.15 (qd, *J* = 12.7 Hz, 4.4 Hz, 2H, Pip 3,5-H), 2.68–2.89 (m, 2H, Pip 2,6-H), 3.53 (tt, *J* = 12.6 Hz, 3.7 Hz, 1H, Pip 4-H), 3.79 (s, 3H, OCH_3_), 3.90 (s, 3H, CH_3_), 4.10–4.44 (m, 2H, Pip 2,6-H), 7.81 (s, 1H, Pyr 3-H). ^13^C NMR (101 MHz, CDCl_3_): δ 28.6 (C(*C*H_3_)_3_), 28.6 (2 × CH_2_, Pip 3,5-C), 34.2 (Pip 4-C), 38.7 (CH_3_), 44.5 (2 × CH_2_, Pip 2,6-C), 51.2 (OCH_3_), 79.8 (*C*(CH_3_)_3_), 111.3 (Pyr 4-C), 141.5 (Pyr 3-C), 148.8 (Pyr 5-C), 154.9 (*C*OOC(CH_3_)_3_), 164.0 (*C*OOCH_3_). ^15^N NMR (41 MHz, CDCl_3_): δ −293.2 (N-Boc), −176.9 (Pyr N-1), −75.1 (Pyr N-2). IR (FT-IR, ν_max_, cm^−1^): 2980, 1705 (C=O), 1688 (C=O), 1234, 779. MS *m/z* (%): 324 ([M + H]^+^, 100%). HRMS (ESI^+^) for C_16_H_25_N_3_NaO_4_ ([M + Na]^+^) calcd 346.1737, found 346.1737.

#### 3.4.9. *tert*-Butyl (3*R*)-3-[4-(methoxycarbonyl)-1-phenyl-1*H*-pyrazol-5-yl]piperidine-1-carboxylate (**5j**)

Compound **3b** was coupled with phenylhydrazine. The obtained residue was purified by column chromatography (SiO_2_, eluent: acetone/*n*-hexane, 1:7, *v*/*v*) to provide compound **5j** as as brownish oil. Yield 379 mg (67%), [α]_D_^20^= 6.4 (*c* 1.12, MeOH). ^1^H-NMR (400 MHz, CDCl_3_): δ 1.40 (s, 10H, C(CH_3_)_3_ and Pip 5-H), 1.63–1.78 (m, 2H, Pip 4,5-H), 2.46 (qd, *J* = 13.1 Hz, 4.1 Hz, 1H, Pip 4-H), 2.74–2.87 (m, 1H, Pip 6-H), 2.88–3.01 (m, 1H, Pip 3-H), 3.54–3.69 (m, 1H, Pip 2-H), 3.86 (s, 3H, OCH_3_), 3.92–4.17 (m, 2H, Pip 2,6-H), 7.36–7.42 (m, 2H, Ph 2,6-H), 7.48–7.54 (m, 3H, Ph 3,4,5-H), 8.05 (s, 1H, Pyr 3-H). ^13^C-NMR (101 MHz, CDCl_3_): δ 25.3 (Pip 5-C), 27.4 (Pip 4-C), 28.5 (C(*C*H_3_)_3_), 36.0 (Pip 3-C), 43.7 (Pip 6-C), 46.1 (Pip 2-C), 51.5 (OCH_3_), 79.6 (*C*(CH_3_)_3_), 112.3 (Pyr 4-C), 126.5 (2 × CH, Ph 2,6-C), 129.4 (Ph 4-C), 129.5 (2 × CH, Ph 3,5-C), 139.0 (Ph 1-C), 143.3 (Pyr 3-C), 148.1 (Pyr 5-C), 154.8 (*C*OOC(CH_3_)_3_), 163.9 (*C*OOCH_3_). ^15^N-NMR (41 MHz, CDCl_3_): δ −159.4 (Pyr N-1), −76.1 (Pyr N-2). IR (FT-IR, ν_max_, cm^−1^): 2975, 1716 (C=O), 1687 (C=O), 1261, 1099, 765. MS *m*/*z* (%): 286 ([M-Boc+H]^+^), 386 ([M + H]^+^), 95%. HRMS (ESI^+^) for C_21_H_27_N_3_NaO_4_ ([M + Na]^+^) calcd 408.1894, found 408.1893.

#### 3.4.10. *tert*-Butyl (3*S*)-3-[4-(methoxycarbonyl)-1-phenyl-1*H*-pyrazol-5-yl]piperidine-1-carboxylate (**5k**)

Compound **3c** was coupled with phenylhydrazine. The obtained residue was purified by column chromatography (SiO_2_, eluent: acetone/*n*-hexane, 1:11, *v*/*v*) to provide compound **5k** as brownish oil. Yield 436 mg (77%), [α]_D_^20^= −6.4 (*c* 0.73, MeOH). ^1^H-NMR (400 MHz, CDCl_3_): 1.40 (s, 10H, C(CH_3_)_3_ and Pip 5-H), 1.60–1.80 (m, 2H, Pip 4,5-H), 2.46 (q, *J* = 13.3 Hz, 1H, Pip 4-H), 2.80 (s, 1H, Pip 6-H), 2.94 (s, 1H, Pip 3-H), 3.48–3.79 (m, 1H, Pip 2-H), 3.86 (s, 3H, OCH_3_), 3.91–4.30 (m, 2H, Pip 2,6-H), 7.32–7.46 (m, 2H, Ph 2,6-H), 7.46–7.56 (m, 3H, Ph 3,4,5-H), 8.05 (s, 1H, Pyr 3-H). ^13^C-NMR (101 MHz, CDCl_3_): δ 25.3 (Pip 5-C), 27.4 (Pip 4-C), 28.5 (C(*C*H_3_)_3_), 36.0 (Pip 3-C), 44.0 (Pip 6-C), 46.0 (Pip 2-C), 51.5 (OCH_3_), 79.6 (*C*(CH_3_)_3_), 112.3 (Pyr 4-C), 126.5 (2 × CH, Ph 2,6-C), 129.4 (Ph 4-C), 129.5 (2 × CH, Ph 3,5-C), 139.1 (Ph 1-C), 143.3 (Pyr 3-C), 148.1 (Pyr 5-C), 154.7 (*C*OOC(CH_3_)_3_), 163.9 (*C*OOCH_3_). IR (FT-IR, ν_max_, cm^−1^): 2979, 1717 (C=O), 1684 (C=O), 1408, 1259, 757. MS *m*/*z* (%): 386 ([M + H]^+^, 96%). HRMS (ESI^+^) for C_21_H_27_N_3_NaO_4_ ([M + Na]^+^) calcd 408.1894, found 408.1892.

#### 3.4.11. *tert*-Butyl (3*R*)-3-[4-(methoxycarbonyl)-1-(4-methylphenyl)-1*H*-pyrazol-5-yl]piperidine-1-carboxylate (**5l**)

Compound **3b** was coupled with *p*-tolylhydrazine hydrochloride. The obtained residue was purified by column chromatography (SiO_2_, eluent: acetone/*n*-hexane, 1:7, *v*/*v*) to provide compound **5l** as brownish oil. Yield 458 mg (78%), [α]_D_^20^= 4.1 (*c* 0.62, MeOH). ^1^H-NMR (400 MHz, CDCl_3_): δ 1.40 (s, 10H, C(CH_3_)_3_ and Pip 5-H), 1.61–1.74 (m, 2H, Pip 4,5-H), 2.43 (s, 4H, Pip 4-H and CH_3_), 2.79 (s, 1H, Pip 6-H), 2.93 (s, 1H, Pip 3-H), 3.48–3.72 (m, 1H, Pip 2-H), 3.85 (s, 3H, OCH_3_), 3.90–4.18 (m, 2H, Pip 2,6-H), 7.22–7.33 (m, 4H, Ph 2,3,5,6-H), 8.03 (s, 1H, Pyr 3-H). ^13^C-NMR (101 MHz, CDCl_3_): δ 21.4 (CH_3_), 25.3 (Pip 5-C), 27.4 (Pip 4-C), 28.5 (C(*C*H_3_)_3_), 36.0 (Pip 3-C), 43.9 (Pip 6-C), 45.8 (Pip 2-C), 51.5 (OCH_3_), 79.5 (*C*(CH_3_)_3_), 112.1 (Pyr 4-C), 126.3 (2 × CH, Ph 2,6-C), 130.0 (2 × CH, Ph 3,5-C), 136.6 (Ph 1-C), 139.6 (Ph 4-C), 143.1 (Pyr 3-C), 148.1 (Pyr 5-C), 154.7 (*C*OOC(CH_3_)_3_), 164.0 (*C*OOCH_3_). ^15^N-NMR (41 MHz, CDCl_3_): δ −157.9 (Pyr N-1), −74.6 (Pyr N-2). IR (FT-IR, ν_max_, cm^−1^): 2976, 1710 (C=O), 1692 (C=O), 1148, 824. MS *m*/*z* (%): 300 ([M-Boc + H]^+^), 400 ([M + H]^+^), 97%. HRMS (ESI^+^) for C_22_H_29_N_3_NaO_4_ ([M + Na]^+^) calcd 422.2050, found 422.2052.

#### 3.4.12. *tert*-Butyl (3*S*)-3-[4-(methoxycarbonyl)-1-(4-methylphenyl)-1*H*-pyrazol-5-yl]piperidine-1-carboxylate (**5m**)

Compound **3c** was coupled with *p*-tolylhydrazine hydrochloride. The obtained residue was purified by column chromatography (SiO_2_, eluent: acetone/*n*-hexane, 1:9, *v*/*v*) to provide compound **5m** as brownish oil. Yield 475 mg (81%), [α]_D_^20^= −4.3 (*c* 0.86, MeOH). ^1^H-NMR (400 MHz, CDCl_3_): δ 1.40 (s, 10H, C(CH_3_)_3_ and Pip 5-H), 1.61–1.77 (m, 2H, Pip 4,5-H), 2.43 (s, 4H, Pip 4-H and CH_3_), 2.79 (s, 1H, Pip 6-H), 2.93 (s, 1H, Pip 3-H), 3.48–3.76 (m, 1H, Pip 2-H), 3.85 (s, 3H, OCH_3_), 3.90–4.26 (m, 2H, Pip 2,6-H), 7.21–7.32 (m, 4H, Ph 2,3,5,6-H), 8.03 (s, 1H, Pyr 3-H). ^13^C-NMR (101 MHz, CDCl_3_): δ 21.4 (CH_3_), 25.3 (Pip 5-C), 27.4 (Pip 4-C), 28.5 (C(*C*H_3_)_3_), 36.1 (Pip 3-C), 43.5 (Pip 6-C), 46.4 (Pip 2-C), 51.5 (OCH_3_), 79.5 (*C*(CH_3_)_3_), 112.1 (Pyr 4-C), 126.3 (2 × CH, Ph 2,6-C), 130.0 (2 × CH, Ph 3,5-C), 136.6 (Ph 1-C), 139.6 (Ph 4-C), 143.1 (Pyr 3-C), 148.1 (Pyr 5-C), 154.7 (*C*OOC(CH_3_)_3_), 164.0 (*C*OOCH_3_). IR (FT-IR, ν_max_, cm^−1^): 2930, 1714 (C=O), 1688 (C=O), 1261, 822. MS *m*/*z* (%): 400 ([M + H]^+^, 95%). HRMS (ESI^+^) for C_22_H_29_N_3_NaO_4_ ([M + Na]^+^) calcd 422.2050, found 422.2051.

#### 3.4.13. *tert*-Butyl (3*R*)-3-{4-(methoxycarbonyl)-1-[3-(trifluoromethyl)phenyl]-1*H*-pyrazol-5-yl}piperidine-1-carboxylate (**5n**)

Compound **3b** was coupled with [3-(trifluoromethyl)phenyl]hydrazine hydrochloride. The obtained residue was purified by column chromatography (SiO_2_, eluent: acetone/*n*-hexane, 1:5, *v*/*v*) to provide compound **5n** as brownish oil. Yield 526 mg (79%), [α]_D_^20^ = 9.9 (*c* 1.31, MeOH). ^1^H-NMR (700 MHz, CDCl_3_): δ 1.39 (br s, 10H, C(CH_3_)_3_ and Pip 5-H), 1.68–1.77 (m, 2H, Pip 4,5-H), 2.50 (qd, *J* = 12.9 Hz, 3.9 Hz, 1H, Pip 4-H), 2.72–3.00 (m, 2H, Pip 3,6-H), 3.52–3.75 (m, 1H, Pip 2-H), 3.87 (s, 3H, OCH_3_), 3.90–4.21 (m, 2H, Pip 2,6-H), 7.52–7.69 (m, 2H, Ph 5,6-H), 7.71 (br s, 1H, Ph 2-H), 7.73–7.82 (m, 1H, Ph 4-H), 8.07 (s, 1H, Pyr 3-H). ^13^C-NMR (176 MHz, CDCl_3_): δ 25.2 (Pip 5-C), 27.3 (Pip 4-C), 28.3 (C(*C*H_3_)_3_), 35.9 (Pip 3-C), 43.9 (Pip 6-C), 45.7 (Pip 2-C), 51.5 (OCH_3_), 79.6 (*C*(CH_3_)_3_), 112.8 (Pyr 4-C), 123.3 (q, *J* = 272.4 Hz, CF_3_), 123.5 (Ph 2-C), 126.1 (Ph 4-C), 129.4 (Ph 5-C), 130.0 (Ph 6-C), 132.2 (Ph 3-C), 139.3 (Ph 1-C), 143.7 (Pyr 3-C), 148.1 (Pyr 5-C), 154.7 (*C*OOC(CH_3_)_3_), 163.5 (*C*OOCH_3_). ^15^N-NMR (41 MHz, CDCl_3_): δ −162.0 (Pyr N-1), −76.1 (Pyr N-2). IR (FT-IR, ν_max_, cm^−1^): 2951, 1717 (C=O), 1688 (C=O), 1130, 1098. MS *m*/*z* (%): 354 ([M-Boc + H]^+^), 454 ([M + H]^+^), 96%. HRMS (ESI^+^) for C_22_H_26_F_3_N_3_NaO_4_ ([M + Na]^+^) calcd 476.1768, found 476.1769.

#### 3.4.14. *tert*-Butyl (3*S*)-3-{4-(methoxycarbonyl)-1-[3-(trifluoromethyl)phenyl]-1*H*-pyrazol-5-yl}piperidine-1-carboxylate (**5o**)

Compound **3c** was coupled with [3-(trifluoromethyl)phenyl]hydrazine hydrochloride. The obtained residue was purified by column chromatography (SiO_2_, eluent: acetone/*n*-hexane, 1:11, *v*/*v*) to provide compound **5o** as brownish oil. Yield 420 mg (63%), [α]_D_^20^ = −9.8 (*c* 0.85, MeOH). ^1^H-NMR (700 MHz, CDCl_3_): δ 1.39 (br s, 10H, C(CH_3_)_3_ and Pip 5-H), 1.68–1.77 (m, 2H, Pip 4,5-H), 2.50 (qd, *J* = 12.9 Hz, 3.9 Hz, 1H, Pip 4-H), 2.72–3.00 (m, 2H, Pip 3,6-H), 3.52–3.75 (m, 1H, Pip 2-H), 3.87 (s, 3H, OCH_3_), 3.90–4.21 (m, 2H, Pip 2,6-H), 7.52–7.69 (m, 2H, Ph 5,6-H), 7.71 (br s, 1H, Ph 2-H), 7.73–7.82 (m, 1H, Ph 4-H), 8.08 (s, 1H, Pyr 3-H). ^13^C-NMR (176 MHz, CDCl_3_): δ 25.2 (Pip 5-C), 27.3 (Pip 4-C), 28.3 (C(*C*H_3_)_3_), 35.9 (Pip 3-C), 43.9 (Pip 6-C), 45.7 (Pip 2-C), 51.5 (OCH_3_), 79.6 (*C*(CH_3_)_3_), 112.8 (Pyr 4-C), 123.3 (q, *J* = 272.4 Hz, CF_3_), 123.5 (Ph 2-C), 126.1 (Ph 4-C), 129.4 (Ph 5-C), 130.0 (Ph 6-C), 132.2 (Ph 3-C), 139.3 (Ph 1-C), 143.7 (Pyr 3-C), 148.1 (Pyr 5-C), 154.7 (*C*OOC(CH_3_)_3_), 163.5 (*C*OOCH_3_). IR (FT-IR, ν_max_, cm^−1^): 2951, 1717 (C=O), 1688 (C=O), 1130, 1099. MS *m*/*z* (%): 454 ([M + H]^+^, 100%). HRMS (ESI^+^) for C_22_H_26_F_3_N_3_NaO_4_ ([M + Na]^+^) calcd 476.1768, found 476.1772.

### 3.5. Synthesis of tert-Butyl 4-[4-(methoxycarbonyl)-1H-pyrazolyl]piperidine-1-carboxylate (***7***)

Compound **3a** (500 mg, 1.5 mmol) was dissolved in EtOH (15 mL) and treated with 55% hydrazine hydrate solution (74 mg, 1.5 mmol). Reaction mixture was stirred at r.t. for 18 h. After removal of the solvent in vacuo, the residue was purified by flash column chromatography (SiO_2_, eluent: acetone/*n*-hexane, 1:7, *v*/*v*) to provide compound **7** as white crystals. Yield 272 mg (60%), mp 128–130 °C. ^1^H-NMR (700 MHz, CDCl_3_): δ 1.47 (s, 9H, C(CH_3_)_3_), 1.64–1.78 (m, 2H, Pip 3,5-H), 1.92–1.99 (m, 2H, Pip 3,5-H), 2.77–2.95 (m, 2H, Pip 2,6-H), 3.53 (t, *J* = 12.0 Hz, 1H, Pip 4-H), 3.83 (s, 3H, OCH_3_), 4.10–4.36 (m, 2H, Pip 2,6-H), 7.96 (s, 1H, Pyr 3(5)-H), 11.52 (s, 1H, Pyr NH). ^13^C-NMR (176 MHz, CDCl_3_): δ 28.5 (C(*C*H_3_)_3_), 30.8 (2 × CH_2_, Pip 3,5-C), 33.8 (Pip 4-C), 44.2 (2 × CH_2_, Pip 2,6-C), 51.2 (OCH_3_), 79.8 (*C*(CH_3_)_3_), 110.1 (Pyr 4-C), 138.7 and 153.6 (Pyr 3(5)-C), 154.9 (*C*OOC(CH_3_)_3_), 164.1 (*C*OOCH_3_). ^15^N-NMR (71 MHz, CDCl_3_): δ −292.7 (N-Boc). IR (FT-IR, ν_max_, cm^−1^): 3208 (N-H), 2980, 1706 (C=O), 1655 (C=O), 1434, 1165, 763. MS *m/z* (%): 210 ([M-Boc + H]^+^) 308 ([M − H]^−^), 97%. HRMS (ESI^+^) for C_15_H_23_N_3_NaO_4_ ([M + Na]^+^) calcd 332.1581, found 332.1581.

### 3.6. Synthesis of tert-Butyl 4-[4-(methoxycarbonyl)-1H-pyrazolyl]piperidine-1-carboxylates (***5i***, ***6i***, ***8***)

A solution of compound **7** (100 mg, 0.3 mmol), KOH (27 mg, 0.5 mmol), and alkyl iodide (1 mmol) in DMF (0.75 mL) was stirred at r.t. for 4 h. The reaction mixture was diluted with EtOAc (10 mL), washed with H_2_O (2 × 15 mL) and brine (15 mL). The organic layer was dried with anhydrous sodium sulfate, filtered, and concentrated under reduced pressure. The crude product was purified by flash chromatography using an eluent—Hex/Me_2_CO in the appropriate ratio.

#### 3.6.1. *tert*-Butyl 4-[4-(methoxycarbonyl)-1-methyl-1*H*-pyrazol-3-yl]piperidine-1-carboxylate (**6i**) and *tert*-Butyl 4-[4-(methoxycarbonyl)-1-methyl-1*H*-pyrazol-5-yl]piperidine-1-carboxylate (**5i**)

Compound **7** was coupled with iodomethane. The obtained residue was purified by column chromatography (SiO_2_, eluent: acetone/*n*-hexane, 1:4, *v*/*v*) to provide an inseparable mixture of regioisomers **6i**:**5i** (5:1) as white crystals. Yield 77 mg (74%). ^1^H-NMR (700 MHz, CDCl_3_) (two isomers are seen in the spectra ratio ~ 5:1 (**6i**:**5i**)): δ 1.44 (s, 9H, C(CH_3_)_3_, (**6i**)), 1.47 (s, 9H, C(CH_3_)_3_, (**5i**)), 1.59–1.76 (m, 2H, Pip 3,5-H, (**6i** and **5i**)), 1.82–1.95 (m, 2H, Pip 3,5-H, (**6i**)), 2.14 (qd, *J* = 12.7 Hz, 4.5 Hz, 2H, Pip 3,5-H, (**5i**)), 2.69–2.95 (m, 2H, Pip 2,6-H, (**6i** and **5i**)), 3.36 (tt, *J* = 11.8 Hz, 3.6 Hz, 1H, Pip 4-H, (**6i**)), 3.54 (t, *J* = 12.6 Hz, 1H, Pip 4-H, (**5i**)), 3.79 (s, 3H, OCH_3_, (**6i**)), 3.80 (s, 3H, OCH_3_, (**5i**)), 3.85 (s, 3H, CH_3_, (**6i**)), 3.92 (s, 3H, CH_3_, (**5i**)), 4.03–4.34 (m, 2H, Pip 2,6-H, (**6i** and **5i**)), 7.78 (s, 1H, Pyr 5-H, (**6i**)), 7.82 (s, 1H, Pyr 3-H, (**5i**)). ^13^C-NMR (176 MHz, CDCl_3_): δ 28.6 (2 × CH_2_, Pip 3,5-C, (**5i**)), 28.6 (C(*C*H_3_)_3_, (**6i** and **5i**)), 31.2 (2 × CH_2_, Pip 3,5-C, (**6i**)), 34.1 (Pip 4-C, (**5i**)), 34.8 (Pip 4-C, (**6i**)), 38.7 (CH_3_, (**5i**)), 39.2 (CH_3_, (**6i**)), 43.8 (2 × CH_2_, Pip 2,6-C, (**6i**)), 44.7 (2 × CH_2_, Pip 2,6-C, (**5i**)), 51.2 (OCH_3_, (**6i**)), 51.2 (OCH_3_, (**5i**)), 79.3 (*C*(CH_3_)_3_, (**6i**)), 79.8 (*C*(CH_3_)_3_, (**5i**)), 110.7 (Pyr 4-C, (**6i**)), 111.3 (Pyr 4-C, (**5i**)), 134.6 (Pyr 5-C, (**6i**)), 141.4 (Pyr 3-C, (**5i**)), 148.7 (Pyr 5-C, (**5i**)), 154.8 (*C*OOC(CH_3_)_3_, (**6i**)), 154.9 (*C*OOC(CH_3_)_3_, (**5i**)), 158.1 (Pyr 3-C, (**6i**)), 163.9 (*C*OOCH_3_, (**6i**)), 164.0 (*C*OOCH_3_, (**5i**)). ^15^N-NMR (71 MHz, CDCl_3_): δ −294.3 (N-Boc, (**5i**)), −183.8 (Pyr N-1, (**6i**)), −178.3 (Pyr N-1, (**5i**)), −77.3 (Pyr N-2, (**6i**)), −76.7 (Pyr N-2, (**5i**)). MS *m*/*z* (%): 324 ([M + H]^+^, 100%). HRMS (ESI^+^) for C_16_H_25_N_3_NaO_4_ ([M + Na]^+^) calcd 346.1737, found 346.1737.

#### 3.6.2. *tert*-Butyl 4-[1-ethyl-4-(methoxycarbonyl)-1*H*-pyrazol-3-yl]piperidine-1-carboxylate (**8**)

Compound **7** was coupled with iodoethane. The obtained residue was purified by column chromatography (SiO_2_, eluent: acetone/*n*-hexane, 1:5, *v*/*v*) to provide compound **8** as white crystals, yield 95 mg (87%), mp 77–78 °C. ^1^H-NMR (700 MHz, CDCl_3_): δ 1.43–1.50 (m, 12H, C(CH_3_)_3_ and CH_2_C*H*_3_), 1.72 (qd, *J* = 12.5 Hz, 4.1 Hz, 2H, Pip 3,5-H), 1.85–1.93 (m, 2H, Pip 3,5-H), 2.80–2.90 (m, 2H, Pip 2,6-H), 3.37 (tt, *J* = 11.8 Hz, 3.6 Hz, 1H, Pip 4-H), 3.79 (s, 3H, OCH_3_), 4.08–4.23 (m, 4H, Pip 2,6-H and C*H*_2_CH_3_), 7.82 (s, 1H, Pyr 5-H). ^13^C-NMR (176 MHz, CDCl_3_): δ 15.2 (CH_2_*C*H_3_), 28.6 (C(*C*H_3_)_3_), 31.1 (2 × CH_2_, Pip 3,5-C), 34.9 (Pip 4-C), 44.4 (2 × CH_2_, Pip 2,6-C), 47.3 (*C*H_2_CH_3_), 51.3 (OCH_3_), 79.9 (*C*(CH_3_)_3_), 110.5 (Pyr 4-C), 133.1 (Pyr 5-C), 155.2 (*C*OOC(CH_3_)_3_), 157.8 (Pyr 3-C), 163.9 (*C*OOCH_3_). ^15^N-NMR (71 MHz, CDCl_3_): δ −168.7 (Pyr N-1), −82.9 (Pyr N-2). IR (FT-IR, ν_max_, cm^−1^): 2980, 1715 (C=O), 1678 (C=O), 1219, 768. MS *m*/*z* (%): 338 ([M + H]^+^, 99%). HRMS (ESI^+^) for C_17_H_27_N_3_NaO_4_ ([M + Na]^+^) calcd 360.1894, found 360.1894.

### 3.7. Synthesis of 5-[1-(tert-Butoxycarbonyl)piperidinyl]-1H-pyrazole-4-carboxylic acids (***9a***–***c***)

Corresponding ester (**5a, 5j, 5k**) (300 mg, 0.78 mmol) was dissolved in MeOH (0.1 mM) and treated with 2 N NaOH (4 equiv). The solution was stirred under reflux for 5 h. After removal of the solvent in vacuo, the residue was dissolved in water (15 mL), washed with EtOAc (2 × 15 mL), acidified with 1 M KHSO_4_ (pH = 1), and washed with EtOAc (2 × 15 mL). The extracts were combined and dried over sodium sulfate, filtered, and concentrated to dryness to give desired compounds which were directly used in the next step without further purification.

#### 3.7.1. 5-[1-(*tert*-Butoxycarbonyl)piperidin-4-yl]-1-phenyl-1*H*-pyrazole-4-carboxylic Acid (**9a**)

Brownish crystals, yield 240 mg (83%), mp 190–192 °C. ^1^H-NMR (400 MHz, CDCl_3_): δ 1.46 (s, 9H, C(CH_3_)_3_), 1.52–1.70 (m, 2H, Pip 3,5-H), 2.26 (qd, *J* = 12.7 Hz, 4.3 Hz, 2H, Pip 3,5-H), 2.48–2.76 (m, 2H, Pip 2,6-H), 3.12 (tt, *J* = 12.5 Hz, 3.6 Hz, 1H, Pip 4-H), 3.97–4.30 (m, 2H, Pip 2,6-H), 7.31–7.38 (m, 2H, Ph 2,6-H), 7.47–7.57 (m, 3H, Ph 3,4,5-H), 8.09 (s, 1H, Pyr 3-H), 9.39 (br s, 1H, OH). ^13^C-NMR (101 MHz, CDCl_3_): δ 28.6 (C(*C*H_3_)_3_), 28.8 (2 × CH_2_, Pip 3,5-C), 35.4 (Pip 4-C), 44.3 (2 × CH_2_, Pip 2,6-C), 79.8 (*C*(CH_3_)_3_), 111.5 (Pyr 4-C), 126.8 (2 × CH, Ph 2,6-C), 129.5 (2 × CH, Ph 3,5-C), 129.7 (Ph 4-C), 139.3 (Ph 1-C), 143.9 (Pyr 3-C), 150.9 (Pyr 5-C), 155.0 (*C*OOC(CH_3_)_3_), 168.4 (COOH). ^15^N-NMR (41 MHz, CDCl_3_): δ −159.6 (Pyr N-1), −75.8 (Pyr N-2). IR (FT-IR, ν_max_, cm^−1^): 2852, 1675 (C=O), 1547, 1424, 764. MS *m*/*z* (%): 370 ([M − H]^−^, 95%). HRMS (ESI^+^) for C_20_H_25_N_3_NaO_4_ ([M + Na]^+^) calcd 394.1737, found 394.1738.

#### 3.7.2. 5-[(3*R*)-1-(*tert*-Butoxycarbonyl)piperidin-3-yl]-1-phenyl-1*H*-pyrazole-4-carboxylic Acid (**9b**)

Brownish crystals, yield 254 mg (88%), mp 86–88 °C, [α]_D_^20^ = 10.4 (*c* 1.10, MeOH). ^1^H-NMR (400 MHz, CDCl_3_): δ 1.41 (s, 10H, C(CH_3_)_3_ and Pip 5-H), 1.60–1.80 (m, 2H, Pip 4,5-H), 2.38–2.52 (m, 1H, Pip 4-H), 2.64–2.86 (m, 1H, Pip 6-H), 2.86–3.13 (m, 1H, Pip 3-H), 3.45–3.75 (m, 1H, Pip 2-H), 3.87–4.23 (m, 2H, Pip 2,6-H), 7.36–7.44 (m, 2H, Ph 2,6-H), 7.49–7.57 (m, 3H, Ph 3,4,5-H), 8.15 (s, 1H, Pyr 3-H). ^13^C-NMR (101 MHz, CDCl_3_): δ 25.2 (Pip 5-C), 27.4 (Pip 4-C), 28.5 (C(*C*H_3_)_3_), 36.1 (Pip 3-C), 43.5 (Pip 6-C), 46.3 (Pip 2-C), 79.7 (*C*(CH_3_)_3_), 111.8 (Pyr 4-C), 126.5 (2 × CH, Ph 2,6-C), 129.6 (3 × CH, Ph 3,4,5-C), 138.9 (Ph 1-C), 144.1 (Pyr 3-C), 148.8 (Pyr 5-C), 154.7 (*C*OOC(CH_3_)_3_), 168.3 (*C*OOH). ^15^N-NMR (41 MHz, CDCl_3_): δ −158.1 (Pyr N-1), −75.9 (Pyr N-2). IR (FT-IR, ν_max_, cm^−1^): 2930, 1686 (C=O), 1412, 1147, 764. MS *m*/*z* (%): 370 ([M − H]^−^, 97%). HRMS (ESI^+^) for C_20_H_25_N_3_NaO_4_ ([M + Na]^+^) calcd 394.1737, found 394.1738.

#### 3.7.3. 5-[(3*S*)-1-(*tert*-Butoxycarbonyl)piperidin-3-yl]-1-phenyl-1*H*-pyrazole-4-carboxylic Acid (**9c**)

Yellowish crystals, yield 243 mg (84%), mp 88–90 °C, [α]_D_^20^ = −10.5 (*c* 1.0, MeOH). ^1^H-NMR (400 MHz, CDCl_3_): δ 1.41 (s, 10H, C(CH_3_)_3_ and Pip 5-H), 1.62–1.78 (m, 2H, Pip 4,5-H), 2.45 (qd, *J* = 12.7 Hz, 7.3 Hz, 1H, Pip 4-H), 2.68–2.87 (m, 1H, Pip 6-H), 2.92–3.08 (m, 1H, Pip 3-H), 3.46–3.76 (m, 1H, Pip 2-H), 3.87–4.21 (m, 2H, Pip 2,6-H), 7.36–7.45 (m, 2H, Ph 2,6-H), 7.47–7.57 (m, 3H, Ph 3,4,5-H), 8.15 (s, 1H, Pyr 3-H). ^13^C-NMR (101 MHz, CDCl_3_): δ 25.2 (Pip 5-C), 27.4 (Pip 4-C), 28.5 (C(*C*H_3_)_3_), 36.1 (Pip 3-C), 43.5 (Pip 6-C), 46.3 (Pip 2-C), 79.7 (*C*(CH_3_)_3_), 111.7 (Pyr 4-C), 126.5 (2 × CH, Ph 2,6-C), 129.6 (3 × CH, Ph 3,4,5-C), 138.9 (Ph 1-C), 144.1 (Pyr 3-C), 148.9 (Pyr 5-C), 154.7 (*C*OOC(CH_3_)_3_), 168.3 (COOH). IR (FT-IR, ν_max_, cm^−1^): 2930, 1687 (C=O), 1412, 1148, 765. MS *m*/*z* (%): 370 ([M − H]^−^, 97%). HRMS (ESI^+^) for C_20_H_25_N_3_NaO_4_ ([M + Na]^+^) calcd 394.1737, found 394.1739.

### 3.8. Synthesis of tert-Butyl 3- and 4-[4-(Phenylcarbamoyl)-1H-pyrazol-5-yl]piperidine-1-carboxylates (***10a***–***c***)

To a solution of the appropriate pyrazole-4-carboxylic acids (**9a**–**c**) (200 mg, 0.54 mmol) and DMAP (7 mg, 0.05 mmol) in DCM (0.1 mM) cooled to 0 °C temperature EDC⋅HCl (114 mg, 0.59 mmol) and aniline (50 mg, 0.54 mmol) were added. The reaction mixture was left at r.t. for 48 h. The solvent was removed under reduced pressure and the crude product was purified by flash chromatography using an eluent—Hex/Me_2_CO (6:1, *v*/*v*).

#### 3.8.1. *tert*-Butyl 4-[1-phenyl-4-(phenylcarbamoyl)-1*H*-pyrazol-5-yl]piperidine-1-carboxylate (**10a**)

White crystals, yield 192 mg (80%), mp 187–189 °C. ^1^H-NMR (400 MHz, CDCl_3_): δ 1.42 (s, 9H, C(CH_3_)_3_), 1.55–1.76 (m, 2H, Pip 3,5-H), 2.18–2.33 (m, 2H, Pip 3,5-H), 2.43–2.71 (m, 2H, Pip 2,6-H), 3.15 (tt, *J* = 12.4 Hz, 3.5 Hz, 1H, Pip 4-H), 3.99–4.24 (m, 2H, Pip 2,6-H), 7.10–7.18 (m, 1H, NHPh 4-H), 7.33–7.39 (m, 4H, NHPh 3,5-H and NPh 2,6-H), 7.48–7.54 (m, 3H, NPh 3,4,5-H), 7.54–7.61 (m, 2H, NHPh 2,6-H), 7.67 (s, 1H, NH), 7.90 (s, 1H, Pyr 3-H). ^13^C-NMR (101 MHz, CDCl_3_): δ 28.6 (C(*C*H_3_)_3_), 29.6 (2 × CH_2_, Pip 3,5-C), 35.4 (Pip 4-C), 44.3 (2 × CH_2_, Pip 2,6-C), 79.6 (*C*(CH_3_)_3_), 116.0 (Pyr 4-C), 120.5 (2 × CH, NHPh 2,6-C), 124.6 (NHPh 4-C), 126.8 (2 × CH, NPh 2,6-C), 129.2 (2 × CH, NHPh 3,5-C), 129.5 (2 × CH, NPh 3,5-C), 129.7 (NPh 4-C), 138.0 (NHPh 1-C), 138.9 (Pyr 3-C), 139.6 (NPh 1-C), 149.2 (Pyr 5-C), 154.9 (*C*OOC(CH_3_)_3_), 161.8 (CONH). ^15^N-NMR (41 MHz, CDCl_3_): δ −252.4 (NH), −159.8 (Pyr N-1), −76.7 (Pyr N-2). IR (FT-IR, ν_max_, cm^−1^): 3390, 1671 (C=O), 1435, 748. MS *m*/*z* (%): 347 ([M-Boc + H]^+^, 99%). HRMS (ESI^+^) for C_26_H_30_N_4_NaO_3_ ([M + Na]^+^) calcd 469.2210, found 469.2209.

#### 3.8.2. *tert*-Butyl (3*R*)-3-[1-phenyl-4-(phenylcarbamoyl)-1*H*-pyrazol-5-yl]piperidine-1-carboxylate (**10b**)

White crystals, yield 197 mg (82%), mp 199–201 °C, [α]_D_^20^ = −20.1 (*c* 0.87, MeOH). ^1^H-NMR (400 MHz, CDCl_3_): δ 1.40 (s, 10H, C(CH_3_)_3_ and Pip 5-H), 1.57–1.70 (m, 1H, Pip 5-H), 1.70–1.83 (m, 1H, Pip 4-H), 2.48 (qd, *J* = 12.9 Hz, 4.0 Hz, 1H, Pip 4-H), 2.69–3.05 (m, 2H, Pip 3,6-H), 3.49–3.81 (m, 1H, Pip 2-H), 3.88–4.24 (m, 2H, Pip 2,6-H), 7.10–7.18 (m, 1H, NHPh 4-H), 7.33–7.45 (m, 4H, NHPh 3,5-H and NPh 2,6-H), 7.46–7.59 (m, 5H, NHPh 2,6-H and NPh 3,4,5-H), 7.72 (s, 1H, NH), 7.92 (s, 1H, Pyr 3-H). ^13^C-NMR (101 MHz, CDCl_3_): δ 25.2 (Pip 5-C), 27.9 (Pip 4-C), 28.5 (C(*C*H_3_)_3_), 36.3 (Pip 3-C), 43.8 (Pip 6-C), 46.6 (Pip 2-C), 79.5 (*C*(CH_3_)_3_), 116.5 (Pyr 4-C), 120.6 (2 × CH, NHPh 2,6-C), 124.7 (NHPh 4-C), 126.5 (2 × CH, NPh 2,6-C), 129.3 (2 × CH, NHPh 3,5-C), 129.5 (3 × CH, NPh 3,4,5-C), 137.9 (NHPh 1-C), 139.1 (NPh 1-C), 139.3 (Pyr 3-C), 147.0 (Pyr 5-C), 154.8 (*C*OOC(CH_3_)_3_), 161.8 (CONH). ^15^N-NMR (41 MHz, CDCl_3_): δ −252.5 (NH), −158.7 (Pyr N-1), −77.1 (Pyr N-2). IR (FT-IR, ν_max_, cm^−1^): 3400 (N-H), 1677 (C=O), 1405, 1137, 751. MS *m*/*z* (%): 447 ([M + H]^+^, 96%). HRMS (ESI^+^) for C_26_H_30_N_4_NaO_3_ ([M + Na]^+^) calcd 469.2210, found 469.2216. The enantiomeric excess was determined by HPLC with a CHIRAL ART Amylose-SA column, t_R_ = 6.5 min (100%), ee = 100%.

#### 3.8.3. *tert*-Butyl (3*S*)-3-[1-phenyl-4-(phenylcarbamoyl)-1*H*-pyrazol-5-yl]piperidine-1-carboxylate (**10c**)

White crystals, yield 161 mg (67%), mp 199–201 °C, [α]_D_^20^ = 19.9 (*c* 0.70, MeOH). ^1^H-NMR (400 MHz, CDCl_3_): δ 1.39 (s, 10H, C(CH_3_)_3_ and Pip 5-H), 1.57–1.67 (m, 1H, Pip 5-H), 1.71–1.80 (m, 1H, Pip 4-H), 2.48 (qd, *J* = 12.8 Hz, 4.0 Hz, 1H, Pip 4-H), 2.69–3.02 (m, 2H, Pip 3,6-H), 3.45–3.83 (m, 1H, Pip 2-H), 3.88–4.20 (m, 2H, Pip 2,6-H), 7.10–7.18 (m, 1H, NHPh 4-H), 7.33–7.44 (m, 4H, NHPh 3,5-H and NPh 2,6-H), 7.46–7.61 (m, 5H, NHPh 2,6-H and NPh 3,4,5-H), 7.75 (s, 1H, NH), 7.92 (s, 1H, Pyr 3-H). ^13^C-NMR (101 MHz, CDCl_3_): δ 25.2 (Pip 5-C), 27.9 (Pip 4-C), 28.5 (C(*C*H_3_)_3_), 36.3 (Pip 3-C), 43.9 (Pip 6-C), 46.6 (Pip 2-C), 79.5 (*C*(CH_3_)_3_), 116.5 (Pyr 4-C), 120.6 (2 × CH, NHPh 2,6-C), 124.6 (NHPh 4-C), 126.5 (2 × CH, NPh 2,6-C), 129.2 (2 × CH, NHPh 3,5-C), 129.5 (3 × CH, NPh 3,4,5-C), 138.0 (NHPh 1-C), 139.1 (NPh 1-C), 139.3 (Pyr 3-C), 147.0 (Pyr 5-C), 154.6 (*C*OOC(CH_3_)_3_), 161.8 (CONH). ^15^N-NMR (41 MHz, CDCl_3_): δ −252.4 (NH), −158.6 (Pyr N-1), −77.0 (Pyr N-2). IR (FT-IR, ν_max_, cm^−1^): 3402 (N-H), 1677 (C=O), 1405, 1137, 751. MS *m*/*z* (%): 347 ([M-Boc + H]^+^), 447 ([M + H]^+^), 99%. HRMS (ESI^+^) for C_26_H_30_N_4_NaO_3_ ([M + Na]^+^) calcd 469.2210, found 469.2210. The enantiomeric excess was determined by HPLC with a CHIRAL ART Amylose-SA column, t_R_ = 6.5 min (1.8% minor enantiomer), t_R_ = 9.2 min (98.2% major enantiomer), ee = 96 %.

## 4. Conclusions

In summary, we developed a new regioselective process for synthesizing 3- or 5- (*N*-Boc-piperidinyl)-1*H*-pyrazole-4-carboxylates as achiral and chiral heterocyclic building blocks. Regioselective synthesis of targeted building blocks was obtained starting from piperidine-4-carboxylic and (*R*)- and (*S*)-piperidine-3-carboxylic acids conversion to the corresponding β-enamino diketones via formation of intermediate β-keto esters. Further investigation of the reaction of β-enamino diketones with various aryl and alkyl hydrazines in various solvents at room temperature proved the regioselective formation of 5-(*N*-Boc-piperidinyl)-1*H*-pyrazole-4-carboxylates in ethanol compared to polar aprotic or nonprotic solvents. Regioisomeric 3-(*N*-Boc-piperidinyl)-1*H*-pyrazole-4-carboxylates were obtained by treating β-enamino diketone with hydrazine hydrate and subsequent alkylation of tautomeric 3(5)-substituted NH-pyrazole with alkylhalides. Furthermore, we demonstrated that 5-(*N*-Boc-piperidinyl)-1*H*-pyrazole-4-carboxylates can be successfully applied to the synthesis of *tert*-butyl 3- and 4-[4-(phenylcarbamoyl)-1*H*-pyrazol-5-yl]piperidine-1-carboxylates by basic hydrolysis and the subsequent reaction of obtained carboxylic acids with aniline in the presence of EDC·HCl and DMAP. The structures of all synthesized compounds were confirmed by detailed NMR spectroscopy and HRMS investigations.

## Data Availability

The data presented in this study are available on request from the corresponding authors.

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
