# Peer review of "Synthesis and Characterization of Novel Methyl (3)5-(N-Boc-piperidinyl)-1H-pyrazole-4-carboxylates"

_molecules, 2021, doi:10.3390/molecules26133808_

Round 1

Reviewer 1 Report

The manuscript

Synthesis and structure of methyl (3)5-(N-Boc-piperidinyl)-1H-pyrazole-4-carboxylates by using nipecotic and isonipecotic acids

 by

Gita Matulevičiūtė 1,2, Eglė Arbačiauskienė 2,*, Neringa Kleizienė 1, Vilija Kederienė 2, Greta Ragaitė 1, Miglė Dagilienė 1, Aurimas Bieliauskas 1, Vaida Milišiūnaitė 1, Frank A. Sløk 3 and Algirdas Šačkus 1,*

is in my opinion quite well written and very accurate in compounds’ characterization. So it is certainly worth of publication on Molecules after minor corrections.

I just have some comments to do, besides to recommending some little corrections.

Lines 15-16: as isonipecotic and nipecotic acids are not so widely known, so it should be better to add the IUPAC name of these acids

Line 50: having specified “5-amino”, also the locant of the carboxylic function should be specified.

Line 57: “their molecules”? maybe molecules therefrom? Or their derivatives?

Line 88 Meldrum acid, EDC HCl, DMAP: it could be better to furnish full names when used for the first time.

Line 100: at least a short comment about the regioselecivity of the reaction should be provided. In particular: do the authors detect only 4a as precursor? Could 6 derive from the nucleophilic attack of the secondary nitrogen atom?

In scheme 3 caption, 6i, 8 and 5i are defined regioisomer. This is not correct. A short comment about total regioselectivity obtained with EtI should be furnished.

Line 272: crude compounds were (not was)

Line 671-704: as far as enantiomers 10b and 10c are concerned, the authors say that 10b was obtained with 100% e.e. and instead 10c with a 96.4% e.e. Now mp is 201.5 for the first and 208.3 for the second. The second, being a stereomeric mixture, should have the lower value, unless other kinds of impurities are present in 10b. can the Authors furnish some comments?

As a general comment, all mp should be furnished as an interval, being the range a very important proof of purity.

Reviewer 2 Report

The publication presented for evaluation (“Synthesis and characterization of novel methyl (3)5-(N-Boc-pi-2 peridinyl)-1H-pyrazole-4-carboxylates”) is very interesting, the authors carried out a very thorough structural analysis, it is a pity that they did not decide to perform any biological tests. The article is written legibly and interestingly. I would just like to know if the authors managed to isolate substances no. 6? Do substances 5 and 6 (respectively 5a and 6a and so on) show differences in the Rf parameter? Are the yields only counted from the 1H NMR spectra?

As for the remarks, I noticed minor shortcomings in the specification of chemical shifts - on the spectra in the supplement, other values in the description were different. Moreover, in the description of the 13C NMR spectra, the assignment should be clarified in some places e. g.:

5a: 28.4 (C (CH3), 79.4 (C (CH3)) - it would have to be somehow marked which is which, e.g. underlines the appropriate C.

Reviewer 3 Report

This manuscript entitled "Synthesis and characterization of novel methyl (3)5-(N-Boc-piperidinyl)-1H-pyrazole-4-carboxylates" describes the synthesis of pyrrole derivatives by condensation reaction. The practical importance of heterocyclic compounds containing chiral amino acids is discussed in the introductory section. The significant biological activities of these compounds are probably the driving force behind the whole work. In the synthetic part, enaminones are first prepared and then condensed to form the final products. The structure of the products is sufficiently determined by 2D NMR techniques. Nevertheless, I do not recommend this work for publication in Molecules for the following reasons:
The preparation of pyrroles by this procedure itself has already been published (ref. 48). The same applies to the preparation of enaminones (ref. 49). The scope of the reaction is limited to the preparation of pyrazoles with the same structural motif by reacting arylhydrazines with enaminones with piperidinyl substituents.
In the present form, I would recommend publication of this article in a more specialized journal, e.g., Journal of Heterocyclic Chemistry or Heterocycles.

Round 2

Reviewer 3 Report

I have no further comments on the manuscript.